# Gram-Scale Synthesis of Graphitic Carbon Nitride Quantum Dots with Ultraviolet Photoluminescence for Fe^3+^ Ion Detection

**DOI:** 10.3390/nano12162804

**Published:** 2022-08-16

**Authors:** Xuemei Lu, Haijun Qin, Jiuzhang Cai, Yuhang Cui, Lixin Liao, Fengzhen Lv, Changming Zhu, Liguang Wang, Jun Liu, Lizhen Long, Wenjie Kong, Fuchi Liu

**Affiliations:** College of Physics and Technology & Guangxi Key Laboratory of Nuclear Physics and Technology, Guangxi Normal University, Guilin 541004, China

**Keywords:** gram-scale preparation, graphitic carbon nitride quantum dots, ultraviolet photoluminescence, Fe^3+^ ion detection

## Abstract

A method for gram-scale synthesis of graphitic carbon nitride quantum dots (g-C_3_N_4_QDs) was developed. The weight of the g-C_3_N_4_QDs was up to 1.32 g in each run with a yield of 66 wt%, and the purity was 99.96 wt%. The results showed that g-C_3_N_4_QDs exhibit a stable and strong ultraviolet photoluminescence at a wavelength of 365 nm. More interestingly, the g-C_3_N_4_QDs can be used as a high-efficiency, sensitive, and selective fluorescent probe to detect Fe^3+^ with a detection limit of 0.259 μM.

## 1. Introduction

The graphitic carbon nitride quantum dots (g-C_3_N_4_QDs) have inspired extensive fundamental and application studies in recent years in the fields of photoelectronic devices [1], ion detection [2], photocatalysis [3], and biological imaging [4] due to their excellent stability, good water-solubility, biocompatibility, low toxicity, and excellent optical properties [5]. In recent years, researchers have carried out numerous works on the preparation of g-C_3_N_4_QDs. The synthesis methods are mainly divided into top-down synthesis and bottom-up synthesis [6]. In the top-down method, for example, Zhang et al. [7] prepared blue, fluorescent g-C_3_N_4_QDs by oxidizing bulk graphitic carbon nitride (g-C_3_N_4_) with a mixture of concentrated H_2_SO_4_ and HNO_3_. By combining a series of processes of acid neutralization, heat treatment, and ultrasonic treatment, Zhan et al. [8] synthesized high water solubility g-C_3_N_4_QDs with blue fluorescence by heating the mixture of g-C_3_N_4_ powder, ethanol, and concentrated KOH solution at 180 °C for 16 h. In the bottom-up method, for example, Liu et al. [9] successfully prepared g-C_3_N_4_QDs with a size of 3–5 nm by heating the mixture of urea and sodium citrate at 180 °C for 1.5 h. However, low yield and purity are still serious problems that will obstruct the widespread application of the g-C_3_N_4_QDs. Using excess organic molecules as precursors makes it especially difficult to separate g-C_3_N_4_QDs from the precursors, and the impurities inevitably remain in g-C_3_N_4_QDs. These difficult-to-remove impurities seriously restricts the accurate understanding of the intrinsic properties of g-C_3_N_4_QDs. Therefore, a simple and effective method for the synthesis of g-C_3_N_4_QDs with a high yield and purity is highly desirable. Although there are various methods to prepare g-C_3_N_4_QDs, the photoluminescence (PL) peak of g-C_3_N_4_QDs is generally located in the blue or green light region [6]. Ultraviolet fluorescence of g-C_3_N_4_QDs is rarely reported. 

Here, we report a gram-scale method for the preparation of g-C_3_N_4_QDs from g-C_3_N_4_. The weight of the g-C_3_N_4_QDs is high; up to 1.32 g in each run. Typically, the yield and purity of the g-C_3_N_4_QDs is up to 66 wt% and 99.96 wt%, respectively. The results show that g-C_3_N_4_QDs exhibit a stable and strong ultraviolet PL at 365 nm with the excitation wavelengths in the range of 220 to 300 nm. More interestingly, the g-C_3_N_4_QDs can be used as a high-efficiency, sensitive, and selective fluorescent probe to detect Fe^3+^ with a detection limit of 0.259 μM. 

## 2. Materials and Methods

### 2.1. The Preparation of g-C_3_N_4_QDs

Bulk g-C_3_N_4_ was synthesized by calcining 10 g of melamine (99.0%, Alfa Aesar, Shanghai, China) at 550 °C for 3 h in a muffle furnace with a heating rate of 0.5 °C·min^−1^ [10,11,12]. The g-C_3_N_4_QDs were prepared from g-C_3_N_4_, refluxed and oxidized, with concentrated nitric acid (HNO_3_, 65–68%) through a self-assembled experimental system [Figure 1a]. In brief, the mixture of 2 g of g-C_3_N_4_ powder and 200 mL of concentrated HNO_3_ were transferred into a 1000 mL round-bottom flask. Then, the mixture was heated at 135 °C in an oil bath followed by refluxing and stirring for 24 h. After cooling to room temperature, the mixture was diluted with deionized water and then evaporated to remove excess HNO_3_ in a rotary evaporator. Then, the obtained mixture was heated at 230 °C for approximately 3 h under a flowing Ar atmosphere (80 mL·min^−1^) to further evaporate HNO_3_, and a small pile of white solid was obtained. Thereafter, the solid was redispersed into deionized water, and then the solution was centrifuged (12,000 rpm) for 15 min to remove unoxidized g-C_3_N_4_ or big particles. The obtained supernatant was diluted and then vacuum filtered by 220 and 25 nm microporous membranes successively. Subsequently, the obtained solution of g-C_3_N_4_QDs was evaporated at 70 °C by rotary evaporators. Finally, the concentrated solution was freeze-dried by a vacuum freeze-drier to obtain 1.32 g of powdered g-C_3_N_4_QDs. 

### 2.2. Fluorescence Detection of Fe^3+^

A volume of 2.0 mL of various concentrations of Fe^3+^ solution (0–100 μM) was added to 2.0 mL of g-C_3_N_4_QDs solutions (0.025, 0.050, 0.075, and 0.100 mg/mL), respectively. The solution was mixed evenly and stood for 1 min to record the fluorescence emission spectra. The relative decrease of PL intensity $\left[(F_{0}-F)/F_{0}\right]$ at excitation wavelength of 247 nm was used for the quantitative analysis, wherein *F*_0_ and *F* are the PL intensities of the g-C_3_N_4_QDs in the absence and presence of Fe^3+^. In order to further verify the applicability of g-C_3_N_4_QDs as fluorescent probes to detect Fe^3+^ in practical applications, the PL response of g-C_3_N_4_QDs towards common cations were tested separately, including K^+^, Na^+^, Ca^2+^, Cd^2+^, Cu^2+^, Fe^2+^, Fe^3+^, Zn^2+^, and Ce^4+^. Selectivity tests were performed using the same procedure as the sensitivity assessment. The concentrations of Fe^3+^ and other metal ions taken were 100 μM.

### 2.3. Characterization

The morphology of each sample was investigated via a transmission electron microscopy (TEM, JEM-2100 F, JEOL, Kariya, Japan) with an acceleration voltage of 200 kV. The content of the chemical composition was measured by X-ray photoelectron spectroscopy (XPS, ESCALAB250, Thermo VG, Waltham, MA, USA) using Al Kα radiation. Fourier transform infrared (FT-IR) spectra were collected on a Brucker TENSOR 27 spectrometer. The absorption spectra were characterized by a UV-vis spectrometer (UV, UV-2700, Shimadzu, Kyoto, Japan) with a wavelength ranging from 200 nm to 800 nm. The impurity content in the sample was measured and analyzed by inductively coupled plasma (ICP, Agilent 720ES, Agilent, Waltham, MA, USA). The PL spectra of the samples were measured by a fluorescence spectrophotometer (PL, RF-5301PC, Shimadzu, Kyoto, Japan).

## 3. Results and Discussion

As shown in Figure 1b, the nearly spherical g-C_3_N_4_QDs were separated from each other and the diameter ranged from 4 to 12 nm. According to the statistics, the average particle size was approximately 7.2 nm. The obtained g-C_3_N_4_QDs were amorphous in structure without detectable lattices and the size was uniform. TEM and HR-TEM images of g-C_3_N_4_ are shown in Appendix A. The crystal structure of g-C_3_N_4_QDs was destroyed due to the introduction of defects. The high disorder of amorphous structure not only makes it have good elastic strain properties in a wide pH range, but also effectively reduces the exciton quenching, thus improving its fluorescence stability [13].

In order to evaluate the purity of g-C_3_N_4_QDs, the impurities’ content in the sample were measured and analyzed by ICP. It can be seen from Table 1 that the g-C_3_N_4_QDs have a low impurity content, and the content of Co, Cu, Mn, and Ni elements is less than 5 ppm. The content of Ca, Fe, and Na elements is 195 ppm, 45 ppm, and 135 ppm, respectively, which are inevitably introduced from deionized water in the preparation process. In general, the purity of g-C_3_N_4_QDs prepared by this work is up to 99.96 wt%.

An XPS analysis is an effective method to understand the chemical composition and elemental chemical states of samples. The XPS survey spectra of the g-C_3_N_4_ and g-C_3_N_4_QDs samples are given in Figure 2a. It can be seen that there are three obvious peaks at the binding energies of 284 eV, 398 eV, and 532 eV of the sample, which correspond to the peaks of the binding energies of C, N, and O elements, respectively [13,14,15]. There are only a few O elements in g-C_3_N_4_, and the atomic percentage of C to N is close to 3:4. The relative content of O element in g-C_3_N_4_QDs is high; up to 26.72 at%, and it has higher oxygen-containing functional group and water solubility than g-C_3_N_4_. This is conducive to the research of ions detection based on g-C_3_N_4_QDs as a fluorescent probe [16]. Compared with g-C_3_N_4_, the content of sp^2^ graphitic carbon and oxygen in the C 1s spectrum of g-C_3_N_4_QDs increased [Figure 2b], indicating that the formation of sp^2^ graphitic carbon was promoted during oxidation and reflux. At the same time, due to the strong oxidizing property of nitric acid, more oxygen-containing functional groups were introduced. As show Figure 2c, the high-resolution N 1s XPS shows three peaks with centers about 398.2 eV, 399.3 eV, and 400.2 eV, corresponding to C-N=C, N-(C)_3_ and C-N-H bonds [17], respectively, indicating that the triazine unit of g-C_3_N_4_ remains in g-C_3_N_4_QDs. However, the ratio of C-N=C content in g-C_3_N_4_QDs decreased, indicating that some triazine units were damaged. The weak peak at about 404.22 eV is attributed to the charge effect or π electron delocalization in the heterocyclic ring [17]. As shown in Figure 2d, high-resolution O 1s XPS further confirmed that oxygen-containing functional groups of g-C_3_N_4_ mainly exist in the form of epoxy group (C-O-C) and carbonyl group (C=O), without carboxyl group (-COOH). The epoxy group (C-O-C) is gradually converted to carbonyl group (C=O) and carboxyl group (-COOH) in the process of oxidation and reflux. The content of each peak points of g-C_3_N_4_ and g-C_3_N_4_QDs can be seen in Appendix A.

As shown in Figure 3a, the FT-IR spectrum of g-C_3_N_4_QDs is mostly like that of g-C_3_N_4_. The characteristic absorption peak at 813 cm^−1^ represented the breathing vibration of tri-s-triazine, which is one typical out-of-plane ring bending vibration mode of g-C_3_N_4_, indicating that the obtained g-C_3_N_4_QDs have the same basic structure as g-C_3_N_4_ [2]. The peak position of 1046 cm^−1^ is mainly caused by the stretching vibration of the C-O bond [18]. The wide peaks in the 1300–1700 cm^−1^ region can be ascribed to the typical stretching vibration modes of C-N heterocycles [19]. The peaks with a wave number of 1388 cm^−1^ and 1731 cm^−1^ correspond to the typical stretching vibration modes of C-N bond and C=N bond, respectively [20]. The peak position of 1780 cm^−1^ is mainly characterized by C=O [21]. The broad absorption band between 3000 cm^−1^ and 3600 cm^−1^ is assigned to O-H stretching vibrations, while the sharp peak is assigned to the tensile vibration of terminal amino (N-H) [22]. These fully indicate that there are several functional groups in the sample, which is consistent with the results of the XPS analysis. Figure 3b shows the UV-vis absorption spectra of g-C_3_N_4_ and g-C_3_N_4_QDs. The main light absorption range of g-C_3_N_4_ and g-C_3_N_4_QDs are in the ultraviolet region. Compared with the bulk g-C_3_N_4_, the edge of g-C_3_N_4_QDs absorption band is blue shifted, which is attributed to the quantum size effect of g-C_3_N_4_QDs [7]. The absorption spectra of g-C_3_N_4_QDs show an obvious absorption peak at 264 nm, which is due to the π-π* electronic transition from HOMO to LUMO of the graphite carbonitride containing tri-s-triazine rings [23]. The UV-vis absorption spectrum can also confirm the existence of tri-s-triazine rings in g-C_3_N_4_QDs. Furthermore, the weak shoulder peak at 365 nm was assigned to the n-π* electronic transition of the C=N and C=O bonds in g-C_3_N_4_QDs [24].

Figure 4a shows the PL spectra of g-C_3_N_4_ at different excitation wavelengths. It can be seen that the emission peak of g-C_3_N_4_ is 435 nm when the excitation wavelength ranges from 220 to 400 nm. Figure 4b shows the PL spectra of g-C_3_N_4_QDs excited at a wavelength of 220–300 nm, and a strong PL emission peak at 365 nm can be seen. It can be found that the PL emission peak of the g-C_3_N_4_QDs is almost unchanged with the change of the excitation wavelength, which indicates that the g-C_3_N_4_QDs have excitation wavelength-independent PL behavior. In addition, the PL intensity of g-C_3_N_4_QDs is almost unchanged under the continuous irradiation by visible light and 365 nm for 12 h [Appendix A, indicating the g-C_3_N_4_QDs have excellent PL performance. The emission peak at 365 nm in the PL spectra is due to the π-π* transition in the unit system of the heterocyclic aromatic hydrocarbon [25], indicating that g-C_3_N_4_QDs still retains the structure of the unit ring of the heterocyclic aromatic hydrocarbon. When 3D g-C_3_N_4_ was broken into 0D g-C_3_N_4_QDs, the emission peak position showed blue shifts. Different from most of the previous g-C_3_N_4_ nanosheets and g-C_3_N_4_QDs with blue or green PL [26], the g-C_3_N_4_QDs in this work have ultraviolet PL.

As shown in Figure 5a–d, the fluorescence quenching intensity of g-C_3_N_4_QDs increases gradually with the increase of Fe^3+^ concentration from 0 to 100 μM. The results showed that g-C_3_N_4_QDs had different responses to Fe^3+^ at different concentrations. The fluorescence of the pristine g-C_3_N_4_QDs (≤0.075 mg/mL) could be completely quenched by the addition of 100 µM Fe^3+^. The Stern-Volmer plots of g-C_3_N_4_QDs at various concentrations for Fe^3+^ has good linearity for concentrations ranging from 0 to10 μM, 0–10 μM, 1–10 μM and 0–10 μM with the following equation: Y_1_ = (0.0544 ± 0.0026 )X_1_ + (0.0415 ± 0.0156)(R^2^ = 0.979), Y_1_ = (0.0745 ± 0.0020)X_1_ + (0.0337 ± 0.0120)(R^2^ = 0.993), Y_1_= (0.0540 ± 0.0048)X_1_ + (0.3010 ± 0.0299)(R^2^ = 0.940) and Y_1_ = (0.0584 ± 0.0023)X_1_ + (0.0315 ± 0.0134)(R^2^ = 0.987), respectively. These results indicated that the fluorescence response of g-C_3_N_4_QDs was best correlated with Fe^3+^ (0–10 μM) when the concentration of g-C_3_N_4_QDs was 0.050 mg/mL. Similarly, the Stern-Volmer plots for the fluorescence quenching of the g-C_3_N_4_QDs with Fe^3+^ has good linearity for concentrations ranging from 10 to 100 μM with the following equation: Y_2_ = (0.0036 ± 0.0003)X_2_ + (0.5880 ± 0.0189)(R^2^ = 0.945), Y_2_ = (0.0024 ± 0.0002)X_2_ + (0.7560 ± 0.0102)(R^2^ = 0.963), Y_2_ = (0.0019 ± 0.0001)X_2_ + (0.7840 ± 0.0069)(R^2^ = 0.972) and Y_2_ = (0.0027 ± 0.0002)X_2_ + (0.6000 ± 0.0127)(R^2^ = 0.973), respectively. These results indicate that the correlation between Fe^3+^ (10–100 μM) and g-C_3_N_4_QDs fluorescence response becomes better with the increase of g-C_3_N_4_QDs concentration. The limit of detection (LOD) is 0.259 μM. The detection limit is based on the Equation (1):(1)$$\mathrm{LOD}=\frac{3\sigma }{k},$$
where σ is the standard deviation of 15 replicate determinations of the blank g-C_3_N_4_QDs (Appendix A); *k* is the slope of the calibration plot [27]. Compared with fluorescent probes based on carbon dots and GQDs (Table 2), the fluorescent probes based on g-C_3_N_4_QDs prepared in this work have obvious advantages in detection limit, reaction time and other analytical characteristics.

In order to further verify the applicability of g-C_3_N_4_QDs as a fluorescent probe to detect Fe^3+^ in practical applications, the PL response of g-C_3_N_4_QDs (0.05 mg/mL) towards common cations (100 μM) was tested separately, including K^+^, Na^+^, Ca^2+^, Cd^2+^, Cu^2+^, Fe^2+^, Fe^3+^, Zn^2+^, and Ce^4+^. As shown in Figure 6a, the fluorescence quenching of g-C_3_N_4_QDs by Fe^3+^ is strong, however the fluorescence quenching of g-C_3_N_4_QDs by other cations is weak. The fluorescence of g-C_3_N_4_QDs is quenched significantly only by Fe^3+^. Figure 6b shows the UV-vis absorption spectrum of common metal ions and the PL excitation and emission spectra of the g-C_3_N_4_QDs. According to the results, the wide range of absorption wavelengths of Fe^3+^ and PL emission peak of g-C_3_N_4_QDs (365 nm) are overlapped, which is very beneficial for fluorescent quenching. The outstanding selectivity and fluorescence quenching of the synthesized g-C_3_N_4_QDs may be ascribed to the absorption of light from g-C_3_N_4_QDs by Fe^3+^. The fluorescence quenching mechanism of the g-C_3_N_4_QDs in the presence of Fe^3+^ was proposed [Figure 6c]. When the excitation wavelength is 247 nm, the emission wavelength of g-C_3_N_4_QDs is 365 nm. In the presence of Fe^3+^, the light emitted by g-C_3_N_4_QDs is absorbed by Fe^3+^, thus causing fluorescence quenching. The excellent performance of the g-C_3_N_4_QDs reflects their potential application as a fluorescent sensor for the quantitative analysis of Fe^3+^ ions in an aqueous solution.

## 4. Conclusions

In summary, we reported a gram-scale method for the preparation of g-C_3_N_4_QDs. The weight of the g-C_3_N_4_QDs is up to 1.32 g in each run by using 2 g of g-C_3_N_4_ as precursor material, with a yield of 66 wt%, and the purity is 99.96 wt%. The results show that g-C_3_N_4_QDs exhibit a stable and strong ultraviolet PL at 365 nm with excitation wavelengths from 220 to 300 nm. More interestingly, using g-C_3_N_4_QDs as a fluorescent probe, Fe^3+^ can be detected within 1 min at the excitation/emission wavelength of 247/365 nm. The PL intensity decreased gradually in the concentration range of 0–100 μM of Fe^3+^. These results indicated that the fluorescence response of g-C_3_N_4_QDs was best correlated with Fe^3+^ (0–10 μM) when the concentration of g-C_3_N_4_QDs was 0.050 mg/mL. The linear equation is Y_1_ = (0.0745 ± 0.0020)X_1_ + (0.0337 ± 0.0120) (R^2^ = 0.993). The correlation between Fe^3+^ (10–100 μM) and g-C_3_N_4_QDs fluorescence response becomes better with the increase of g-C_3_N_4_QDs concentration. The best linear equation is Y_2_ = (0.0027 ± 0.0002)X_2_ + (0.6000 ± 0.0127) (R^2^ = 0.973), and the limit of detection is 0.259 μM. The high-efficiency, sensitive, and selective detection of Fe^3+^ was realized. All the results suggest that the obtained g-C_3_N_4_QDs with high UV fluorescence have potential applications in the fields of photoelectronic devices and ion detection.

## Figures and Tables

**Figure 1 nanomaterials-12-02804-f001:**
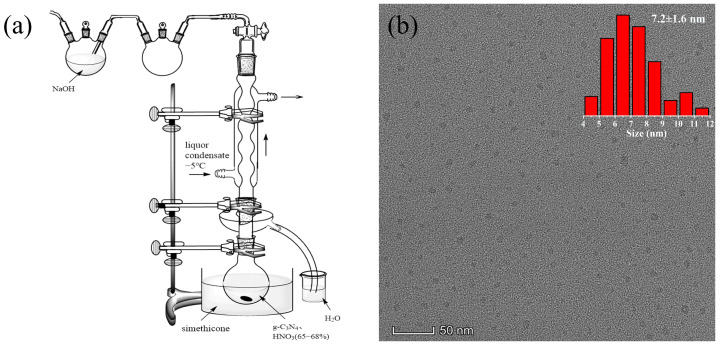
(**a**) Schematic diagram of the self-assembled experimental system. (**b**) TEM image and the particle size map of g-C_3_N_4_QDs.

**Figure 2 nanomaterials-12-02804-f002:**
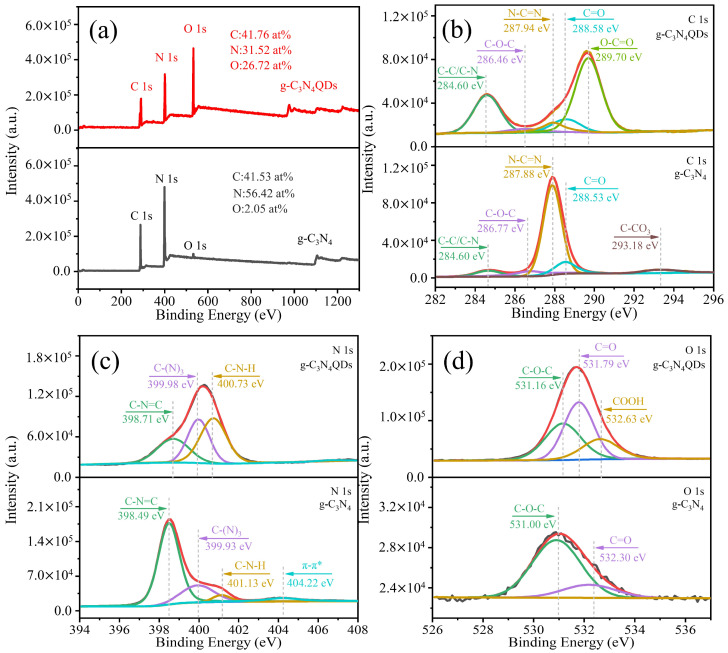
XPS survey spectra (**a**) and the high-resolution XPS spectra of (**b**) C 1s, (**c**) N 1s, and (**d**) O 1s of g-C_3_N_4_ and g-C_3_N_4_QDs.

**Figure 3 nanomaterials-12-02804-f003:**
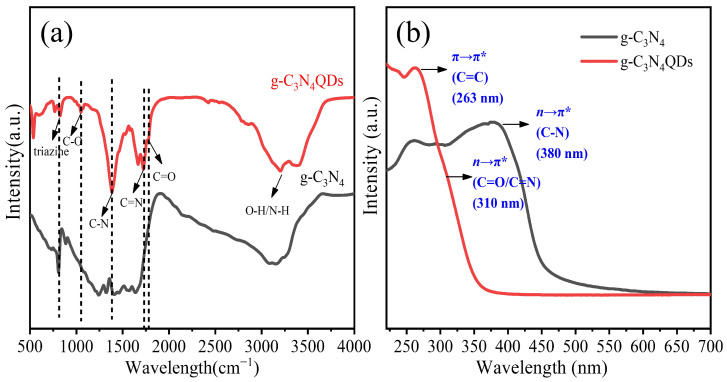
FT-IR spectra of (**a**) and UV-vis absorption spectra of (**b**) of g-C_3_N_4_ and g-C_3_N_4_QDs.

**Figure 4 nanomaterials-12-02804-f004:**
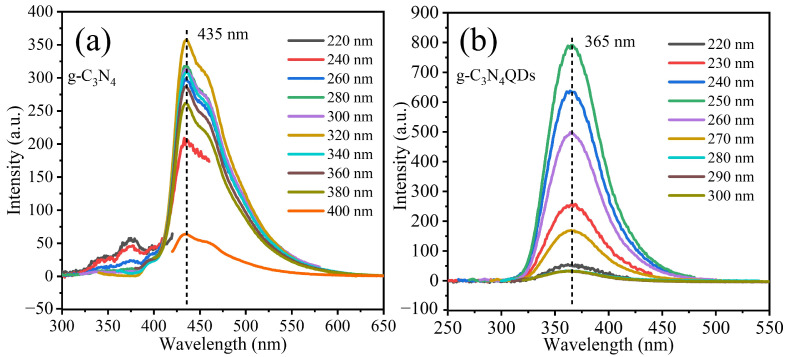
PL spectra of g-C_3_N_4_ (**a**) and g-C_3_N_4_QDs (**b**).

**Figure 5 nanomaterials-12-02804-f005:**
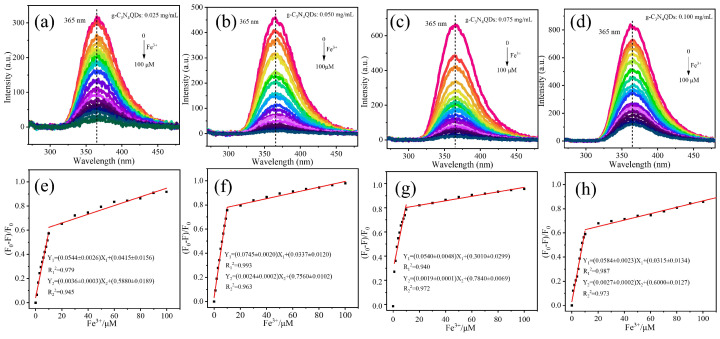
(**a**–**d**) Fluorescence responses of different concentrations of g-C_3_N_4_QDs in the presence of different concentrations of Fe^3+^ (excitation wavelength, 247 nm). (**e**–**h**) The relationship between the fluorescence quenched ratio and the concentration of Fe^3+^ from 0 to 100 μM.

**Figure 6 nanomaterials-12-02804-f006:**
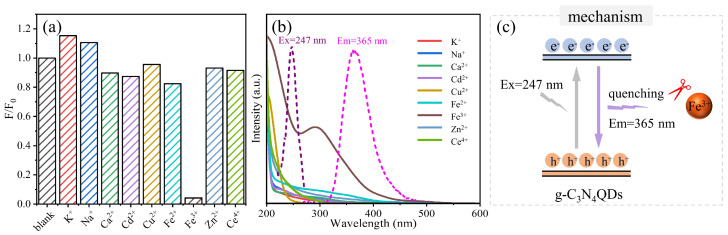
(**a**) PL response of the g-C_3_N_4_QDs (0.050 mg/mL) in the solution of common metal ions (100 μM). (**b**) UV-vis absorption spectrum of common metal ions and the PL excitation and emission spectra of the g-C_3_N_4_QDs. (**c**) Proposed fluorescence quenching mechanism of the g-C_3_N_4_QDs in the presence of Fe^3+^.

**Table 1 nanomaterials-12-02804-t001:** Inductively coupled plasma (ICP) analysis data of g-C_3_N_4_QDs.

Samples (ppm)	Ca	Co	Cu	Fe	Mn	Na	Ni
g-C_3_N_4_QDs	195	<5	<5	45	<5	135	<5

**Table 2 nanomaterials-12-02804-t002:** Based fluorescent probes for Fe^3+^ detection.

Materials	Linear Range (μmol·L^−1^)	Detection Limit (μmol·L^−1^)	Reaction Time (min)	Ref
Carbon dots	0–20	0.32	10	[28]
g-CNQDs	2–200	1	2	[25]
N-CQDs	3.32–32.26	0.7462	2	[29]
S-GQDs	0–0.7	0.0042	10	[30]
N and S doped Carbon dots	6.0–200	0.8	2	[31]
C-dots	12.5–100	9.97	-	[32]
g-C_3_N_4_QDs	0–100	0.259	1	This work

## Data Availability

All data generated and analyzed during this study are included in this article and the attached Appendix A.

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
