# Peer review of "Gram-Scale Synthesis of Graphitic Carbon Nitride Quantum Dots with Ultraviolet Photoluminescence for Fe3+ Ion Detection"

_nanomaterials, 2022, doi:10.3390/nano12162804_

Round 1

Reviewer 1 Report

The XPS high resolution spectra deconvolution is quite odd since no C-N bonds are reported in the analysis of C1s for the QD while the FTIR show higher peaks for these species. It will be necessary to reformulate to accommodate the peak of the CN bonds (or at least also report them if superimposition with C-O groups happen). Please report the intensities of the high-resolution spectra.

Please shows intensities values in figure 4.

Table 2 should also include data for g-C3N4 and explain which the advantages of using g-C3N4QD instead of g-C3N4.

Reviewer 2 Report

The article by Liu et al is devoted to a new technique for obtaining gram amounts of carbon nitride nanoparticles and testing it as a luminescent sensor. In general, the ms is detailed and interesting, but the authors need to pay more attention to the mathematical processing of data. Any measurements should always be given with accurately calculated measurement errors. Without them, the mathematical and analytical meaning of the parameters is completely lost.

1. Page 2, lines 42, 43. The accuracy with which the masses and yields are found is clearly excessive, it is better to indicate 1.32 g and 66%.

2. P2L75 Sn4+ and Pb4+ ions due to limited propensity for hydrolysis and a strongly oxidizing properties of Pb(IV)

3. P4L117 The XPS method does not provide a reliable and reproducible determination of oxygen up to a hundredth of a fraction, so the entered value of 26.72 seems to be erroneous.

4. P7L188-197 The given parameters of the Stern-Volmer equation are too precise. They should be considered as Y=(k±error)x+(b±error), reducing the number of significant digits, taking into account standard rounding rules.

Round 2

Reviewer 1 Report

The authors' corrections have improved the manuscript. I have no further suggestions. I recommend its publication.